# A Crusade Throughout the World’s Oceans: Genetic Evidence of the Southern Bluefin Tuna *Thunnus maccoyii* and the Pacific Bluefin Tuna *Thunnus orientalis* in Brazilian Waters

**DOI:** 10.3390/biology14040340

**Published:** 2025-03-26

**Authors:** Rafael Schroeder, Rodrigo Sant’Ana, André O. S. Lima, Juliana A. Dallabona, Gabriela S. Delabary, Lucas Gavazzoni, Luciana de Oliveira, Yan de O. Laaf, Paulo Travassos

**Affiliations:** 1Laboratório de Estudos Marinhos Aplicados, Escola Politécnica, Universidade do Vale do Itajaí (UNIVALI), Rua Uruguai 458, Itajaí 88302-901, Brazil; rsantana@univali.br (R.S.); gavazzonilucas@gmail.com (L.G.); 2Centro Interdisciplinar de Investigação Marinha e Ambiental (CIIMAR), Terminal de Cruzeiros do Porto de Leixões, Avenida General Norton de Matos S/N, 4550-208 Matosinhos, Portugal; 3Laboratório de Genética Molecular, Escola Politécnica, Universidade do Vale do Itajaí (UNIVALI), Rua Uruguai 458, Itajaí 88302-901, Brazil; lima@univali.br (A.O.S.L.); julianaa.dallabona@gmail.com (J.A.D.); gabrielad@univali.br (G.S.D.); lucybiotec@gmail.com (L.d.O.); yan.laaf@hotmail.com (Y.d.O.L.); 4Laboratório de Ecologia Marinha, Universidade Federal Rural de Pernambuco (UFRPE), Rua Dom Manuel de Medeiros S/N, Recife 52171-900, Brazil; paulo.travassos@ufrpe.br

**Keywords:** zoogeography, species range extension, bluefin tuna, *Thunnus orientalis*, *Thunnus maccoyii*, North Pacific, Indic Ocean, Southwest Atlantic

## Abstract

This study highlights the presence of large pelagic species in distant marine ecosystems in the waters of southeast–south Brazil. Important commercial species were identified, such as Southern bluefin tuna (*Thunnus maccoyii*) and Pacific bluefin tuna (*Thunnus orientalis*). The presence of the cold-water *T. maccoyii* and the warm-water *T. orientalis* suggests a long migration from common species ranges that could be influenced by climate change. These results imply a readjustment of the spatial management structures for these species.

## 1. Introduction

Knowledge of the spatial distribution of species is a fundamental requirement for managing and conserving marine biodiversity [1,2,3]. Pelagic longline fishing off the Brazilian coast has low selectivity, and includes target species, such as highly migratory pelagic fish (e.g., *Thunnus* spp. and swordfish *Xiphias gladius*) but also a considerable list of bony fish and elasmobranchs as accompanying fauna [4], including some species of high commercial value, such as the Atlantic bluefin tuna *Thunnus thynnus* [5,6,7]. Some of these species, however, are difficult to identify when they are landed in ports, so accurate methodologies need to be used to accurately identify the organisms caught, such as mitochondrial genetic analysis [8,9,10].

In the southeast–south region of Brazil, the Yellowfin tuna *Thunnus albacares*, Albacore *Thunnus alalunga*, and Bigeye tuna *Thunnus obesus*, along with the Blackfin tuna *Thunnus atlanticus,* are frequently found in longline catches, with other tuna and tuna-like species occurring in smaller volumes [11]. These species have a widely known distribution [12,13,14,15] and are managed by the International Commission for the Conservation of Atlantic Tuna (ICAAT). In recent years, however, relatively frequent catches of a temperate-water tuna species have been recorded, referred to as the Southern bluefin tuna (*Thunnus maccoyii*) [11]. These recent appearances of species of important commercial value prompted a study investigating the catch composition of these larger, commercially important specimens, using genetic markers [16].

The Southern bluefin tuna occur in southern seas, off the Antarctic continent, mainly between the latitudes of 30° and 50° S to nearly 60° S [17], with a thermal preference for water between 19 and 21 °C, but can withstand much colder (3 °C) and warm temperatures (30 °C) during the spawning season [18]. Despite its predicted distribution in Brazilian waters, its participation in pelagic longline catches remains unknown. In addition, this species is managed by the Commission for the Conservation of Southern Bluefin Yuna (CCSBT) and already faces intense fishing efforts. As indicated by the CCSBT *T. maccoyii* stock is overfished, but overfishing is not currently occurring [19].

Another extraordinary record involves a species of tuna not yet known in the Atlantic [20], the Pacific bluefin tuna *Thunnus orientalis*, which is widely distributed in the Pacific Ocean and seasonally inhabits subarctic, temperate, and tropical waters in the North Pacific Ocean and temperate waters in the Southern Hemisphere around Australia and New Zealand, marginally entering the eastern Indian Ocean [21,22,23,24,25]. The species is evaluated by the International Scientific Committee for Tuna and Tuna-like Species in the North Pacific Ocean (ISC) and placed as Near Threatened on the International Union for Conservation of Nature (IUCN) Red List Category and Criteria. The current population trend is decreasing [25].

The fact that population studies involving complementary methodologies, such as morphological, molecular and otolith chemistry analyses and tag-return data, show that both species of bluefin tuna consist of panmictic stocks for *T. maccoyii* [17,26,27] and for *T. orientalis* [28,29,30], and their occurrence in the South Atlantic may increase fishing mortality and spatial overlap of management units with implications for the management of these species [31]. Thus, this study aimed to identify tuna specimens, especially concerning possible occurrences of southern bluefin tuna caught by the longline fleet based in southern Brazil, through the genetic analysis of samples collected. The paper reports recent astonishing records of tuna species known previously from the North Pacific in locations in the Southwest Atlantic. The study hereby discusses how these might have appeared thousands of kilometers from their previously known species ranges.

## 2. Materials and Methods

### 2.1. Origin of Samples and Fishing Data

The Industrial Fisheries Monitoring Program (PMAP) conducted by UNIVALI/LEMA has been carrying out analyses of commercial catches and biological samples in southeast and south Brazil since 2000 [11]. In more recent years (2018–2022), data included landed catches, fishing efforts, and areas of operation of 5869 fishing trips conducted by vessels operating pelagic longlines (Figure 1). A total of 39 species have been identified from the landed catch, together with some bluefin tuna of unknown identification. By 2020 and 2022, biological tissue samples had been collected from specimens of 10 large-sized bluefin tuna, commercially caught. Samples were collected from distinct specimens that could not be identified onboard. The samples were stored in alcohol and sent to the Laboratory of Molecular Genetics (LMG) at the University of Vale do Itajaí, where they were stored in a freezer (−20 °C) for further processing.

### 2.2. DNA Extraction and COI Gene Amplification and Sequencing

Genomic DNA was extracted from muscle tissue samples using the DNeasy Blood and Tissue Kit (Qiagen, Hilden, Germany) according to the manufacturer’s instructions. After extraction, both the DNA quality and concentration were assessed simultaneously by measuring the absorbance ratio at 260 nm/280 nm using 2 µL of the sample on the NanoQuant Plate (Tecan, Männedorf, Switzerland) with a Tecan Infinite^®^ 200 Pro microplate reader (Tecan, Männedorf, Switzerland). The absorbance ratio confirmed the absence of significant protein contamination, ensuring that the DNA purity was suitable for downstream applications.

The DNA samples were then subjected to polymerase chain reaction (PCR) to amplify the cytochrome c oxidase I (COI) gene, a mitochondrial marker commonly used for species identification. The PCR followed the protocol by [32], utilizing the primer pair VF2_t1 (5′-TGTAAAACGACGGCCAGTCAACCAACCACAAAGACATTGGCAC-3′) and FR1d_t1 (5′-CAGGAAACAGCTATGACACCTCAGGGTGTCCGAARA AYCARAA-3′), both of which included an M13 tail to facilitate sequencing. The PCR reaction mixture (25 µL total volume) consisted of 50 ng of genomic DNA, 1x Taq DNA polymerase buffer (Sigma-Aldrich, St. Louis, MO, USA), 0.2 mM of each dNTP, 0.5 µM of each primer, 2 mM MgCl_2_, 1 U Taq DNA polymerase (Sigma-Aldrich, USA), and ultrapure water to reach the final reaction volume.

PCR amplification was conducted using the Veriti^®^ Thermal Cycler (Applied Biosystems, San Francisco, CA, USA) with the following cycling conditions: an initial denaturation at 94 °C for 2 min, followed by 35 cycles of 94 °C for 30 s, 52 °C for 40 s, and 72 °C for 1 min, with a final extension at 72 °C for 10 min. The success of the PCR amplification was confirmed by agarose gel electrophoresis. A 1% agarose gel was prepared, and PCR products were mixed with loading buffer and stained with Diamond Nucleic Acid Dye (Promega, Madison, WI, USA). The amplified DNA fragments, approximately 700 base pairs in size, were visualized using a UV transilluminator, with their molecular weight determined against a λ Hind III marker. Electrophoresis was performed at 50 V and 120 mA for 90 min, and the gels were documented using the EOS Utility 2 software (v2.14.20.0, Canon, Japan). The amplified COI genes from the ten (10) samples were sequenced to confirm species identification. Sequencing was performed using the Sanger chain termination method on the Applied Biosystems ABI 3500 X Genetic Analyzer, producing sequences of approximately 700 base pairs. The sequencing was conducted by GoGenetic (Curitiba, PR, Brazil), ensuring accurate and reliable identification of the amplified products.

### 2.3. Sequence Processing, Identification, and Phylogenetic Analysis

Following sequencing, the raw reads from the amplified COI genes of the nine (9) samples were processed using CLC Genomics Workbench (v6.7, CLC bio, Aarhus, Denmark). Each sample was sequenced twice, generating reads from both the forward and reverse primers. The initial step in data processing involved trimming the terminal sequences based on a quality threshold of Q_30_ to ensure high accuracy.

The trimmed sequences from both directions were then aligned and merged to create a consensus sequence for each sample. These consensus sequences were saved in FASTA format for subsequent analyses. Each FASTA file was uploaded to the Barcode of Life Data System (BOLD) for species identification and compared against the Species Level Barcode Records database [33]. This extensive database includes 5,040,349 COI sequences representing 246,329 species and 120,909 interim species. By comparing our sequences to this comprehensive reference, we achieved precise species identification based on the highest similarity matches with known entries in the BOLD database.

The consensus sequences obtained were further analyzed to explore phylogenetic relationships. All consensus sequences were aligned using MEGA 11 (Molecular Evolutionary Genetics Analysis, v. 11) software. This alignment process ensured that the sequences shared a common start and end point, allowing for uniform comparison across samples. To enhance the phylogenetic analysis, additional sequences were retrieved from the BOLD database, given its higher level of curation and consistency with the species identification step. Specifically, mitochondrial DNA sequences from all *Thunnus* species were downloaded from BOLD, along with *Katsuwonus pelamis* sequences, which were used as an outgroup. A custom script was employed to extract between 5 and 10 sequences per *Thunnus* species, focusing on those that matched the COI gene region obtained from our samples. The combined dataset, including both the new sequences from this study and the reference sequences from BOLD, was aligned using the ClustalW algorithm in MEGA. This comprehensive alignment formed the basis for the subsequent phylogenetic analysis. The phylogenetic relationships were inferred using the Maximum Likelihood (ML) method, employing the Kimura 2-parameter model with a gamma distribution (G) to account for rate variation among sites [34]. The number of discrete gamma categories was set to 5, and the robustness of the phylogeny was tested with 1000 bootstrap replicates. The initial tree was automatically generated using the Nearest-Neighbor-Interchange (NNI) heuristic method. The final phylogenetic tree was carefully analyzed to interpret the evolutionary relationships among the *Thunnus* species represented in the dataset.

## 3. Results

### 3.1. DNA Extraction and COI Amplification

DNA was successfully extracted from all fish muscle samples, with concentrations ranging from 21.95 ng/µL to 179.5 ng/µL and 260 nm/280 nm absorbance ratios between 1.96 and 2.19, indicating high purity suitable for downstream analyses (Table 1). PCR amplification of the COI gene was also successful for all samples, producing amplicons of approximately 720 base pairs. The expected band sizes confirmed successful amplification in each case. The amplified COI sequences were processed and compared against the Barcode of Life Data System (BOLD) database for species identification. The identification results showed four species within the *Thunnus* genus. Six samples were identified as *Thunnus maccoyii*, two samples were identified as *Thunnus obesus*, one sample identified as *Thunnus albacares* with 100% identity, and one sample matched *Thunnus orientalis* with 99.85% identity (Table 1). These results confirm the efficiency of DNA extraction, amplification, and sequencing processes, reflecting the reliability of species identification through the BOLD database.

### 3.2. Phylogenetic Analysis

The phylogenetic relationships among eight *Thunnus* species were inferred using the Maximum Likelihood (ML) method, based on the Kimura 2-parameter model with a gamma distribution (G) to account for rate variation among sites. The analysis was performed using MEGA software, with 1000 bootstrap replicates to assess the robustness of the tree. The sequences obtained were aligned with reference sequences of *Thunnus* species from the Barcode of Life Data System (BOLD), and *Katsuwonus pelamis* were used as an outgroup to root the tree. The resulting phylogenetic tree (Figure 2) shows a well-supported division between the species within the *Thunnus* genus. The six samples identified as *Thunnus maccoyii* are grouped within a highly supported clade with a bootstrap value of 77%. This clade is distinct from other *Thunnus* species, confirming the strong genetic differentiation of *Thunnus maccoyii* from its relatives.

The *Thunnus obesus* samples also form a well-defined clade with high bootstrap support (66%), demonstrating clear genetic consistency among the individuals within this species. Interestingly, the *Thunnus obesus* clade is closely related to the *Thunnus atlanticus*, *Thunnus albacares* and *Thunnus tonggol* clade, which is supported by a bootstrap value of 21%. This suggests a closer evolutionary relationship between these two species compared to others in the tree. The single sample identified as *Thunnus orientalis* forms a distinct clade alongside reference sequences from BOLD, with a bootstrap value (57%). The proximity of the *Thunnus orientalis* clade to the *Thunnus alalunga* group suggests a potential evolutionary connection between these species, although further investigation would be needed to clarify the exact relationship.

The *Thunnus albacares* sample is placed within its expected clade, though the bootstrap support for this group is relatively low (23%), suggesting that the evolutionary relationships among members of this species may be less clearly resolved with the current dataset. However, the sample still clusters with high confidence within the *Thunnus albacares* species group, supporting its identification. The outgroup, *Katsuwonus pelamis*, is separated from the *Thunnus* genus, confirming its more distant evolutionary relationship. This separation provides a solid rooting point for the tree and further validates the observed phylogenetic structure within *Thunnus*. Overall, the tree supports the identification results obtained through BOLD and provides a clear visualization of the evolutionary relationships among the *Thunnus* species analyzed in this study. The bootstrap values generally indicate strong support for the major clades, particularly for *Thunnus obesus*, *Thunnus maccoyii*, and *Thunnus orientalis*.

### 3.3. Occurrence of Thunnus orientalis and T. maccoyii in Brazilian Waters

The geographical distribution of *Thunnus orientalis* outside the typical species habitat observed in other studies suggests that the specimen observed in the current study have migrated to Brazilian waters, passing through the Indic Ocean (Figure 3). Details of how this specimen might have appeared several kilometers from the previously known species range are discussed.

## 4. Discussion

The current study confirms the presence of the Pacific bluefin tuna, *T. orientalis*, in Brazilian waters. Analysis of the partial CO1 sequence from the Pacific bluefin tuna caught off Southern Brazil using BLAST (v2.14.0, NCBI, Bethesda, MD, USA) search showed 99.85% similarity, respectively, with published reference sequences for *T. orientalis* (BOLD accession no: FOA884-04). In addition, results for *T. maccoyii*, *T. obesus* and *T. albacares* were complete achieving 100% similarity, respectively, with published reference sequences for *T. maccoyii* captured off the West of Rottnest Island in Western Australia (BOLD accession no: FOA877-04); for *T. obesus* from South Africa (BOLD accession no: SAFCO26-11); and for *T. albacares,* without a record from the area of capture (GenBank accession no: ANGBF42001-19).

The phylogenetic tree inferred from the Maximum Likelihood method using the Kimura 2-parameter model showed that the obtained CO1 sequence analysis from the South Brazil large-size tuna specimens showed high accuracy in distinguishing among tuna species, namely *T. albacares*, *T. obesus*, *T. maccoyii*, and *T. orientalis,* as observed in similar studies [35,36,37]. On the other hand, the differentiation of *T. alalunga* into two clusters could be attributed to distinct populations of the species, as demonstrated by mt-DNA and microsatellite analyses [8,38,39]. According to the partial CO1 sequence, the upper cluster is referent to samples collected from the Atlantic (BOLD accession no: NEELS228-14; FOA868-04; GBMND68190-21; MOBIL8868-18; NEELS313-14) and the lower part from samples collected in the Mediterranean Ocean (BOLD accession no: DNATR1700-13; DNATR1703-13; DNATR1704-13). Despite some studies showing that the efficiency of CO1 as a marker for differentiating tuna species was previously considered problematic [8], the high accuracy provided by 1000 times bootstrap obtained in the sequences can be interpreted as valid for species identification. The COI gene is a reliable marker for species identification due to its high interspecific variation and intraspecific conservation [40]. Studies indicate that COI is highly effective in identifying *Thunnus* spp. [41]. Its standardized use in the BOLD database ensures the global comparability of sequences [42].

**Figure 3 biology-14-00340-f003:**
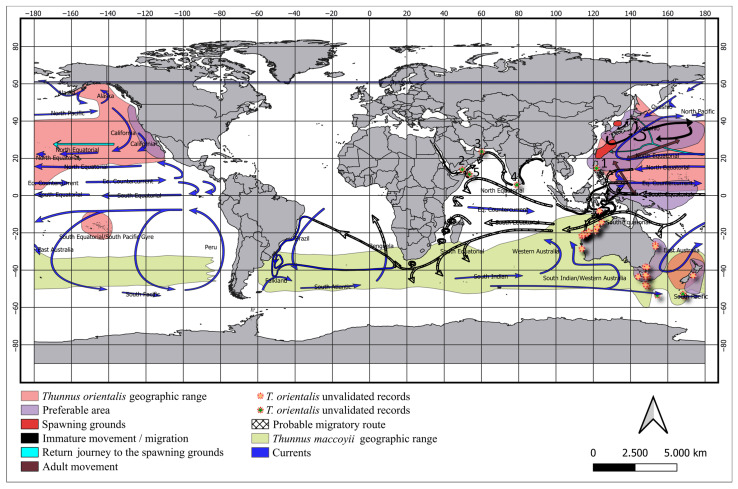
Spatial distribution of the biological samples (colored circles) collected by the pelagic longline fleet on the southeast–south Brazilian coast between 2018 and 2022. The colored areas represented the geographic range of Southern bluefin tuna *Thunnus maccoyii* and the Pacific bluefin tuna *Thunnus orientalis*, preferable areas, and spawning grounds. The arrows represent the movements of immature and mature individuals of *T. orientalis* and marine currents. Stars represent unvalidated and validated records of *T. orientalis* obtained in the FishBase Dataset [43] and scientific papers [9,10,37]. The dashed arrows represent a probable migratory route for *T. orientalis* until arriving in Brazilian waters.

Besides the differentiation in the phylogenetic tree, *T. orientalis* was the only species without a predicted distribution for Brazilian waters [12,13]. The movements of *T. orientalis* are among the best documented of any highly migratory species, according to the ISC, even considering the large inter-annual variations of movement (numbers of migrants, the timing of migration, and migration routes) [44,45,46,47]. Adult fish of *T. orientalis* spawn in the Western North Pacific Ocean between the Philippine Sea and the Ryukyu (Nansei) Islands from late April to early July [24,48,49] and in the coastal area of the Sea of Japan from July to August [50,51]. After spawning, they generally migrate north to feeding grounds in the western Pacific and may reside there for their whole lives, being known as residents [49]. Young fish aged 0–1 hatched in the waters around the Ryukyu Islands and eastern Taiwan are transported north with the Kuroshio Current in the summer as they grow, whereas age-0 fish hatched in the Sea of Japan migrate along the Japanese and Korean coasts [21,52,53]. An unknown portion of immature *T. orientalis* ages 1–3 in the western Pacific underwent eastward migration across the Pacific Ocean, as demonstrated by analysis of electronic archival tags [21,45,54] and stable isotope in muscle tissues [30,55] and otoliths [56,57,58]. This migratory portion enters the California Current Large Marine Ecosystem at juvenile or subadult stages for feeding, spending up to four years before returning to the western Pacific for forage and reproduction, which depends on the ocean conditions [30,44,52,59]. A mechanism of eastward trans-Pacific migration is hypothesized due to the limitation of food sources in the western Pacific and favorable oceanographic conditions [60]. While *T. orientalis* are in the eastern Pacific, the juveniles make seasonal north–south migrations along the west coast of North America [45,61]. In the spring, *T. orientalis* reside in the waters off the southern coast of Baja California and, as the water warms up in summer, *T. orientalis* move northwest into the southern California bight. By fall, *T. orientalis* is found in the waters of central and northern California. After spending 3–4 years in the eastern Pacific, *T. orientalis* moves westward, presumably for purposes of spawning, as no spawning ground has been observed outside of the western Pacific. This westward migration was observed from December to March as *T. orientalis* began their migration along the coast of California [45].

In the West Pacific, the spawning areas are located between Taiwan and the Philippine Sea, the latter being considered as the main spawning area [49,53,62]. After spawning, *T. orientalis* should typically migrate north during the spring. In the springtime, high primary production and favorable environmental conditions are observed in the south portion of the Ryukyu Islands and the Philippines between April and June, along with similar conditions in the Indian Ocean, close to the coast of Timor-Leste, Java, and Sumatra [45,46,63]. Despite the southward migration of mature fish not being fully understood, the existence of favorable environmental conditions in both oceans suggests a possible connection between them, as well as the search for suitable habitats during the springtime, following the habitat suitability model’s predictions [64]. The most important oceanographic feature present in this area is the Indonesian Throughflow (ITF), which could serve as a possible passage for *T. orientalis* between the Pacific and the Indic oceans [65,66,67]. This flow could potentially allow vagrants of *T. orientalis* to enter other parts of the Indian Ocean [9,10,37]. The trajectory through the ITF can also be supported by some unvalidated records of species occurrence that could not be strictly validated for morphological or genetic analysis [20,67].

The first valid record was found in the doorsteps of the ITF pathway through the Philippines Sea provided by a “bluefin tuna” caught on 16 May North of Polilio Island by a hook and line fishing vessel. The individual was captured at 15°23.570 N, 121°59.868 E, and its identity was confirmed by molecular analysis as *Thunnus orientalis*, with 100% similarity for partial CO1 sequence (BOLD accession no: KU058179.1 [37]). In the Indian Ocean, the first valid record of *T. orientalis* was represented by a male bluefin tuna of 250 cm TL, weighting 217 kg, fished by a longline on 11 May 2017, between the Arabian Sea and the Sea of Oman, in the Sultanate of Oman. In this case, the morphometric characteristics, the meristic characters, and particularly the genetic analyses (mt-DNA) combined, confirmed the individual as Pacific bluefin tuna, the first found in this part of the Indian Ocean [9]. A second “probable” *T. orientalis* was captured by a longline near the Sri Lanka waters, in May of 2020. Despite the records never having been officially published, available pictures revealed a 2 m total length fish, close to 200 kg. Considering the location of the catch and images from caudal keels, this specimen was another vagrant Pacific bluefin tuna [10]. On 14 May 2021, two Yemenite fishermen caught a large bluefin tuna (227.5 cm total length and 375 kg) using handlines some miles offshore. Due to high prices on the local market, the fish was rapidly commercialized, making it impossible to obtain tissues for molecular analysis. However, the external characteristics, size, and weight were also compatible with Pacific bluefin tuna; in this case, the fish was caught around mid-May [10].

What all these three specimens of *T. orientalis* had in common is that they were all mature specimens and were captured in May, and the approximate reported location is in the outflow of the Equatorial Current, encircling most of the known distribution of *T. orientalis*. The approximate east–west trans-Pacific migration partially overlaps the route of the Pacific North Equatorial Current, which passes into the two known spawning areas of *T. orientalis*. Near the west Pacific coast, this current bifurcates to the north, giving rise to the Kuroshio Current. In the westward direction, it boosts the ITF, which imports water with a higher temperature and lower salinity from the western equatorial Pacific Ocean. The ITF also receives the contribution from the Pacific South Equatorial Current. When the ITF flows through Lombok Strait, Ombai, and the Timor Passages and enters the Indian Ocean, it transports a large amount of relatively warm and fresh water into the Indian Ocean towards Africa within the Indian Equatorial Current [64,65,66].

Following the validated (and non-validated) records of the occurrence of *T. orientalis* in the Indian Ocean [9,10,37,43], all follow the course of the ITF in a northerly direction originating from the North Equatorial Current towards Sri Lanka [64,65,66]. The most likely route, in line with validated records of the species’ occurrence, heads north with the Indian North Equatorial Current, aligning with the record of a *T. orientalis* caught in Sri Lanka, in May 2020 [37] and a second adult specimen caught in Sur, Sultanate of Oman, on 11 May 2017 in the Arabian Sea [9]. The other two positively identified fish were found in an area quite close by, in the Gulf of Aden [10]. Further west, there is the example of the adult specimen caught in Ash Shihr, Yemen, on 14 May 2021 and a juvenile specimen caught off Cape Guardafui, Somalia, between September and March in the years 1966–1972 [10]. This specimen had represented the end of the line of distance for migratory vagrant specimens of *T. orientalis* until the appearance of the individual captured off Brazilian waters.

The long journey to eastern Brazilian waters presents a huge extension when compared to the most distant ever fifth specimen recorded off Cape Guardafui, Somalia. From Cape Guardafui onwards, the Brazilian *T. orientalis* possibly took transportation along the Agulhas Current, which originates from the extension of the Indian North Equatorial Current flowing to South Africa [67]. After crossing the confluence of the Agulhas, the specimen could migrate as far as east Brazilian waters through the Atlantic South Equatorial Current. However, given the vastness of the Brazilian waters and the variability in tuna populations, this sample size may be too small to accurately represent the overall catch composition and the occurrence of different tuna species. It would be beneficial to increase the sample size in future studies to enhance the reliability of the results. For example, a larger sample could provide more accurate estimates of the proportion of each tuna species in the catches.

The current work suggests a possible migratory route for *T. orientalis* based on the evidence presented. Other authors have suggested an antipodal link between the North Pacific and the Atlantic Ocean through the Drake Passage, following the Upper Circumpolar Deep Water [68]. The proposed migratory route for *T. orientalis* to Brazilian waters is based on limited evidence and some assumptions. The evaluation of the present hypothesis would require more data, such as additional genetic analysis of specimens along the proposed route or direct tracking data (e.g., from satellite tags) and otolith chemical records. Another possible effect that could be correlated with this long migration involves climate change [10]. These marine environmental modifications logically affect tuna in the Atlantic, Indian, and Pacific Oceans [69,70,71]. The many and complex effects of climate change in the Indian Ocean in association with these increasing occasional catches may represent the beginning of a wider distribution range, as previously suggested [10,72]. Historic genetic analysis (mt-DNA) indicated that a differentiation event may have occurred in *T. maccoyii;* however, a recent connectivity between the two oceans may have connected *T. maccoyii* populations [73]. Moreover, the genetic structure analysis of *T. maccoyii* and *T. orientalis* provided evidence of their population expansion dating to the middle Pleistocene [72]. The validation of this hypothesis would, therefore, imply a readjustment of spatial management structures for these species [31].

## 5. Conclusions

The current study allowed us to identify for the first time in Brazil the southern bluefin tuna (*Thunnus maccoyii*) and the Pacific bluefin tuna (*Thunnus orientalis*), caught by the longline fleet operating in southeast and south Brazil, through DNA analysis. As mentioned above, records of catches of the species had already been reported in recent years, but unofficially and without much certainty about the process of identifying these catches due to the difficulties imposed by the dynamics of landing the catches in ports. Thus, there is no doubt that the species is caught with some frequency by the pelagic longline fishing fleet for tuna and tuna-like fish that operates in southeast and south Brazil. Although both tuna species are not under the responsibility of ICCAT for conservation and fisheries management, a task that falls to the CCSBT and ICS, respectively, these catches must be officially recorded and duly communicated to Brazil’s fisheries management body (SAP/MAPA), which will be responsible for transmitting this data to ICCAT, which will inform the CCSBT and the ICS.

## Figures and Tables

**Figure 1 biology-14-00340-f001:**
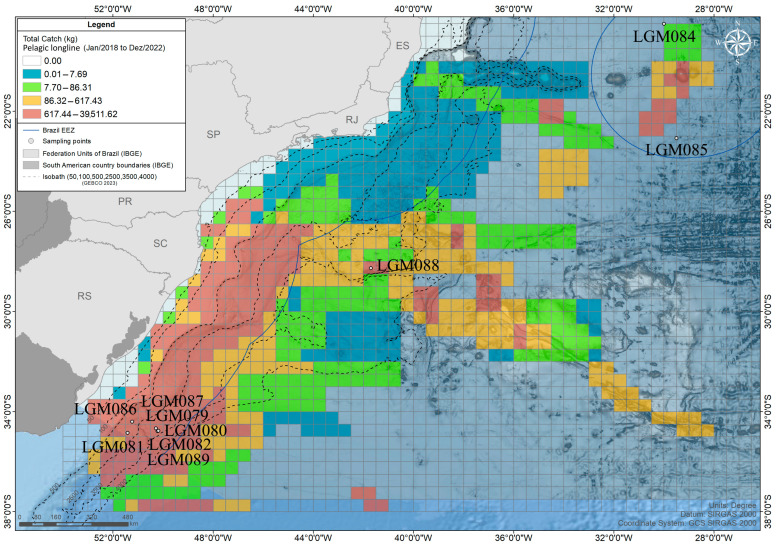
Spatial distribution of the operations of the pelagic longline fleet along the Brazilian coast in 2018–2022 and volumes of total catch in the major fishing grounds: Espírito Santo (ES), Rio de Janeiro (RJ), São Paulo (SP), Paraná (PR), Santa Catarina (SC), and Rio Grande do Sul (RS). The white spots represent the geographical position of the samples individualized by a unique Sample ID.

**Figure 2 biology-14-00340-f002:**
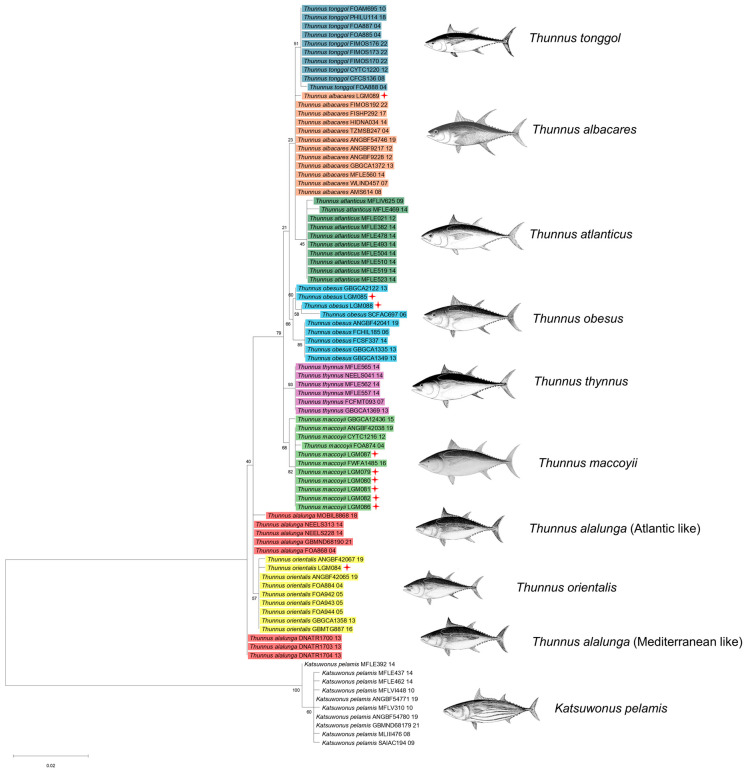
Phylogenetic tree inferred using the Maximum Likelihood (ML) method based on the Kimura 2-parameter model with gamma distribution. The sequences obtained in this study are indicated by filled stars (✦). Bootstrap values (1000 replicates) are shown next to the branches. *Katsuwonus pelamis* was used as an outgroup. Illustrations of the tuna species were adapted from FAO Species Catalogue: Vol. 2 Scombrids of the World and represented by different colors.

**Table 1 biology-14-00340-t001:** DNA concentration, quality (260 nm/280 nm), and species identification results for nine *Thunnus* samples based on the Barcode of Life Data System (BOLD). Species identity (%) reflects the match between the COI sequences obtained and the reference sequences in the BOLD database, along with their corresponding reference codes. Accession numbers of sequences deposited in the GenBank database were also included.

Sample ID	DNA Analysis		BOLD Best ID			Accession Numbers
	Concentration (ng/uL)	Quality (260/280 nm)	Species	ID (%)	Ref	
LGM079	125.6	2.05	*Thunnus maccoyii*	100	FOA877-04	PV265537
LGM080	53.4	1.96	*Thunnus maccoyii*	100	FOA877-04	PV265538
LGM081	71.3	2.00	*Thunnus maccoyii*	100	FOA877-04	PV265539
LGM082	55.1	2.11	*Thunnus maccoyii*	100	FOA877-04	PV265540
LGM084	179.5	2.02	*Thunnus orientalis*	99.85	FOA884-04	PV265541
LGM085	27.7	2.09	*Thunnus obesus*	100	SAFC026-11	PV265542
LGM086	21.9	2.19	*Thunnus maccoyii*	100	FOA877-04	PV265533
LGM087	49.1	2.00	*Thunnus maccoyii*	100	FOA877-04	PV265534
LGM088	80.7	2.06	*Thunnus obesus*	100	SAFC026-11	PV265535
LGM089	42.7	2.19	*Thunnus albacares*	100	ANGBF42001-19	PV265536

## Data Availability

Data will be made available upon reasonable request.

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
