# Peer review of "A Crusade Throughout the World’s Oceans: Genetic Evidence of the Southern Bluefin Tuna Thunnus maccoyii and the Pacific Bluefin Tuna Thunnus orientalis in Brazilian Waters"

_biology, 2025, doi:10.3390/biology14040340_

Round 1

Reviewer 1 Report

Comments and Suggestions for Authors

This article presents a study of commercially exploited and threatened tuna (Thunnus) species in Brazilian waters. Data are presented on the spatial distribution of catches of the pelagic longline fleet in the south-east-south coast of Brazil, in which four species of tunas are common. In addition to the latter, the authors' genetic study confirmed the presence of warm-water Pacific T. maccoyii and cold-water subantarctic T. orientalis in commercial catches. Despite the fact that the number of samples is only 10, the results are quite objective. Phylogenetic tree for 8 Thunnus, and spatial distribution of two species with migratory pathways are given. The authors suggest that one of the reasons for the change in distribution may be the influence of climate change. The article is suitable for the special issue of J Biology and can be accepted for publication with minor changes (see comments).

Author Response

L18. bluefin tuna. There must be uniformity along the whole text.

  1. The text was standardized. See revised MS L3, 25, 26, 55, 64, 94.

L88-98. Sampling method. What was a general principle to select specimens for tissue sampling? Unusual specimens? Fishes of unknown species? Preliminary identification as maccoyii? Explain please.

  1. Suggestion accepted. The figure label was modified. See revised MS L96-97.

L140-141 Spatial distribution of the operatins of the pelagic longline fleet in the southeast–south Brazilian coast between 2018 and 2022 in the major fishing grounds: Spatial distribution of the operations of the pelagic longline fleet along Brazilian coast in 2018-2022 and volumes of total catch. The major fishing grounds:

  1. Suggestion accepted. The figure label was modified. See revised MS L140-143.

L181. The identification results showed three species within the Thunnus genus. But four species listed afterwards (lines 182-184): maccoyii, orientalis, obesus, and albacares.

  1. The text was corrected. See revised MS L182.

L183-184 and two samples were identified as Thunnus obesus and Thunnus albacares [it means by one each? But see Table 1], two samples were identified as Thunnus obesus and [omitted?:] one as Thunnus albacares [as listed in Table 1].

  1. The text was corrected. See revised MS L182-185.

L188-191 Table 1. Explain somewhere, why there are 9 samples in table 1 but not 10.

  1. The text was corrected. See revised MS L189-192.

L193. The phylogenetic relationships among Thunnus species. The phylogenetic relationships among eight Thunnus species.

  1. Suggestion accepted. See revised MS L194.

L200. The seven samples identified as Thunnus maccoyii are grouped within a highly supported. But six maccoyii samples are marked by filled stars on the tree

  1. The text was corrected. See revised MS L201.

L262. Fig3. The figure is over-crowded by signs, arrows and colors.

Suggestions: 1) remove colored circles for the T albacares and T obesus from Fig 3 [since the fig is titled for two other species; on my opinion is better to designate all four on Fig 1]. 2) Reduce some number of current arrows. 3) Mark the currents with a light blue color, which is associated with the waters and makes it easier to perceive and understand (and possibly, give in other design); 4) Stars in the legend, representing unvalidated and validated records of T. orientalis, are shown by one and the same color. Make them by two different colors if possible (if not, give just one star in the legend for Fish Base Dataset and literature {since they are of same color on the map }). Move authors 1,2,3,4,5 from the legend into the Figure subscription; pay attention that [9, 10] in legend are not listed in Fig.5-subscription. Name the Fig. 3 something like this: Spatial distribution of Thunnus maccoyii and Thunnus orientalis, with preferable areas, spawning grounds, and migrations shown for the latter.

  1. Suggestions accepted. See the new Figure 3 in the revised MS.

L341. Nera the west Pacific coast t. Near the west Pacific

  1. The text was corrected. See revised MS L345.

Reviewer 2 Report

Comments and Suggestions for Authors

The manuscript presents an interesting and novel finding regarding the presence of Southern Bluefin tuna (Thunnus maccoyii) and Pacific Bluefin tuna (Thunnus orientalis) in Brazilian waters, which are not typically found in this region. The study uses genetic evidence to support these findings and explores potential migratory routes and implications for fisheries management. The research is well-conducted, and the results are significant for understanding the distribution dynamics of these species, especially in the context of climate change.

  1. The study sampled only 10 large - size Bluefin - tuna specimens. Given the vastness of the Brazilian waters and the variability in tuna populations, this sample size may be too small to accurately represent the overall catch composition and the occurrence of different tuna species. It would be beneficial to increase the sample size in future studies to enhance the reliability of the results. For example, a larger sample could provide more accurate estimates of the proportion of each tuna species in the catches.
  2. The proposed migratory route for Thunnus orientalis to Brazilian waters is based on limited evidence and a series of assumptions. While the authors suggest a path through the Indian Ocean and along ocean currents, more data, such as additional genetic analysis of specimens along the proposed route or direct tracking data (e.g., from satellite tags), are needed to confirm this hypothesis. Without stronger evidence, the proposed migratory route remains speculative.
  3. In the phylogenetic analysis, the bootstrap support for some clades, such as the Thunnus albacares clade, is relatively low (19%). This may indicate that the current dataset and analysis methods are not sufficient to clearly resolve the evolutionary relationships within this species. The authors could consider adding more genetic markers or increasing the number of sequences in the analysis to improve the resolution of the phylogenetic tree and better understand the relationships among tuna species.
  4. The rationale for using the COI gene for species identification could be further justified, especially in light of potential limitations or alternative markers.
  5. The study reports the presence of Thunnus orientalis and Thunnus maccoyii in Brazilian waters, but the geographical and temporal context of these findings could be better contextualized. For example, are these occurrences isolated events, or do they represent a trend? The authors should discuss whether similar findings have been reported in nearby regions or if these are the first documented cases. 

Author Response

  1. The study sampled only 10 large - size Bluefin - tuna specimens. Given the vastness of the Brazilian waters and the variability in tuna populations, this sample size may be too small to accurately represent the overall catch composition and the occurrence of different tuna species. It would be beneficial to increase the sample size in future studies to enhance the reliability of the results. For example, a larger sample could provide more accurate estimates of the proportion of each tuna species in the catches.
  1. Suggestion accepted. See revised MS L370-375.

  1. The proposed migratory route for Thunnus orientalis to Brazilian waters is based on limited evidence and a series of assumptions. While the authors suggest a path through the Indian Ocean and along ocean currents, more data, such as additional genetic analysis of specimens along the proposed route or direct tracking data (e.g., from satellite tags), are needed to confirm this hypothesis. Without stronger evidence, the proposed migratory route remains speculative.
  1. Suggestion accepted. See revised MS L379-383.

  1. In the phylogenetic analysis, the bootstrap support for some clades, such as the Thunnus albacares clade, is relatively low (19%). This may indicate that the current dataset and analysis methods are not sufficient to clearly resolve the evolutionary relationships within this species. The authors could consider adding more genetic markers or increasing the number of sequences in the analysis to improve the resolution of the phylogenetic tree and better understand the relationships among tuna species.
  1. Suggestion accepted. We were able to improve the support value for the T. albacares branch by 4%, following suggestions from the reviewer, although for this time we were not able to add other genetic markers. Several tests were made and the sequences ANGBF41809-19, ANGBF54732-19 and GBGCA1370-13 were removed, and more specific sequences were added: FIMOS192-22, FISHP292-17, HIDNA034-14, TZMSB247-04, and AMS614-08. The same methodology and parameters for the tree were applied. See the new Figure 2.

  1. The rationale for using the COI gene for species identification could be further justified, especially in light of potential limitations or alternative markers.
  1. Suggestion accepted. See revised MS L262-265.
  2. The study reports the presence of Thunnus orientalis and Thunnus maccoyii in Brazilian waters, but the geographical and temporal context of these findings could be better contextualized. For example, are these occurrences isolated events, or do they represent a trend? The authors should discuss whether similar findings have been reported in nearby regions or if these are the first documented cases. 
  3. Suggestion accepted. See revised MS L388-392.